# The α-synuclein PET tracer [18F] ACI-12589 distinguishes multiple system atrophy from other neurodegenerative diseases

Ruben Smith [1,2,15], Francesca Capotosti [3,15], Martin Schain[1,4,5], Tomas Ohlsson[6], Efthymia Vokali[3], Jerome Molette [3], Tanja Touilloux[3], Valerie Hliva [3], Ioannis K. Dimitrakopoulos[3], Andreas Puschmann[2,7,8], Jonas Jögi[9], Per Svenningsson [10], Mattias Andréasson[10], Christine Sandiego[11], David S. Russell [11], Patricia Miranda-Azpiazu[12], Christer Halldin[12], Erik Stomrud[1,13], Sara Hall [1,13], Klas Bratteby [6], Elina Tampio L'Estrade[6], Ruth Luthi-Carter[3], Andrea Pfeifer[3], Marie Kosco-Vilbois[3], Johannes Streffer [3,14] ✉ & Oskar Hansson [1,13] ✉

A positron emission tomography (PET) tracer detecting α-synuclein pathology will improve the diagnosis, and ultimately the treatment of α-synuclein-related diseases. Here we show that the PET ligand, [18F]ACI-12589, displays good in vitro affinity and specificity for pathological α-synuclein in tissues from patients with different α-synuclein-related disorders including Parkinson's disease (PD) and Multiple-System Atrophy (MSA) using autoradiography and radiobinding techniques. In the initial clinical evaluation we include 23 participants with α-synuclein related disorders, 11 with other neurodegenerative disorders and eight controls. In vivo [18F]ACI-12589 demonstrates clear binding in the cerebellar white matter and middle cerebellar peduncles of MSA patients, regions known to be highly affected by α-synuclein pathology, but shows limited binding in PD. The binding statistically separates MSA patients from healthy controls and subjects with other neurodegenerative disorders, including other synucleinopathies. Our results indicate that α-synuclein pathology in MSA can be identified using [18F]ACI-12589 PET imaging, potentially improving the diagnostic work-up of MSA and allowing for detection of drug target engagement in vivo of novel α-synuclein targeting therapies.

Neurodegenerative diseases share the common feature of pathologic protein aggregation, with β-amyloid (Aβ), tau and α-synuclein (α-syn) being the most prevalent aggregating proteins. While significant progress has been made for the diagnosis of Aβ and tau, no reliable imaging biomarker has so far been available for α-syn[1].

The most prominent α-syn-related disorders, Parkinson's disease (PD), dementia with Lewy bodies (DLB), and multiple system atrophy (MSA), show distinct clinical manifestations corresponding to the differences in regional distributions of the underlying α-syn pathology.

Clinically, PD is characterized by rigidity, bradykinesia and resting tremor. DLB shares these symptoms of parkinsonism, appearing in parallel with cognitive impairment characterized by hallucinations and fluctuating executive and visuospatial cognitive abnormalities[1,2]. MSA is characterized by autonomic failure in combination with parkinsonism and/or ataxia and frequently progresses to involve brain stem symptoms such as dysarthria and swallowing difficulties[3]. MSA can present with a phenotype dominated by parkinsonism (MSA-P) or one dominated by cerebellar ataxia (MSA-C), but a mix of the two

phenotypes is also common. Diagnosis of MSA can be difficult, particularly in early disease stages, because this relies mainly on clinical symptoms[4] that are shared with other movement disorders such as PD, progressive supranuclear palsy (PSP) and corticobasal degeneration. Thus, tools to aid in the early differential diagnosis of MSA would comprise a highly significant advance.

α-syn-containing deposits are the histopathologic hallmark of synucleinopathies, but α-syn accumulation is nonetheless heterogeneous in different diseases in terms of its distribution, conformation and seeding capacity. In PD, the pathology is mainly present as neuronal inclusions in the form of Lewy bodies or neurites, typically appearing first in the brainstem, spreading via the midbrain/substantia nigra to the medial temporal cortex, and via the mesocortex to the neocortex[5]. DLB presents a cellular pathology similar to that of PD, but with a wider distribution in cortical regions[5]. In MSA, the distinguishing α-syn pathology manifests primarily in oligodendrocytes as glial cytoplasmic inclusions, which are especially prominent in the white matter of the brainstem and cerebellum[5,6]. Differences between the synucleinopathies at the ultrastructural and functional levels are also observed. High-resolution cryo-electron microscopy (cryo-EM) reveals that the folding structures of fibrillar α-syn demonstrate differences in PD and DLB compared to MSA[7,8]. Moreover, recently established cyclic amplification techniques (e.g. RT-QuIC) also demonstrate differences in α-syn seeding capacities of cerebrospinal fluid (CSF) from patients with PD versus MSA[1,8,9].

Positron emission tomography (PET) radiotracers have been successfully developed for pathological species of Aβ and tau, providing unprecedented insights in correlating the developing molecular pathology in the living brain with evolving clinical symptoms. In addition, Aβ PET imaging has recently been used as a single primary surrogate efficacy measure in an Alzheimer's disease immunotherapy trial[10]. Furthermore, tau tracers have shown their capabilities not only to detect tau inclusions in the brain but also to support differential diagnosis based on signal distribution[11–13].

Thus, our first aim was to leverage AC Immune's proprietary Morphomer® library to identify a brain-penetrant small molecule with high affinity for α-syn aggregates and good selectivity over other potential brain pathologies (or co-pathologies). Secondly, we needed to determine whether the selected candidate α-syn radiotracer, [18F]ACI-12589, could provide a robust and meaningful signal to reveal disease-specific synuclein neuropathology in vivo.

Here, we present [18F]ACI-12589 PET data from healthy control subjects and patients with α-synucleinopathies and other neurological diagnoses, showing a specific and reproducible retention pattern in patients with MSA. The radiotracer [18F]ACI-12589 may therefore constitute an important diagnostic tool to identify, characterize and track α-syn pathology in MSA clinically. As such, this can be considered a long-awaited breakthrough in state biomarkers for α-synucleinopathies.

## Results
### Preclinical characterization
ACI-12589 is a small molecular weight compound identified from the screening of AC Immune's Morphomer® platform based on its α-synuclein-binding properties. The structure of the tracer [18F]ACI-12589 and its precursor, ACI-15051, as well as the radiolabeling reaction, are shown in Supplementary Fig. 1.

Target engagement of [3H]ACI-12589 was evaluated using autoradiography in brain sections from one monogenic PD case carrying a G51D mutation in the α-synuclein (SNCA) gene and one MSA case. [3H]ACI-12589 gave a clear signal (Fig. 1a; top panel) overlapping with α-syn neuropathology visualized using anti-pS129 α-syn immunohistochemistry (IHC) (Fig. 1a; bottom panel). The specificity of the pathological α-syn signal was confirmed by the extent of co-localization with pS129 immunofluorescence as well as

its displacement by an excess of unlabeled ACI-12589 (Fig. 1a; middle panel).

Saturation binding experiments performed by autoradiography showed dissociation constants ($K_d$) of 17 nM for a familial PD case and 28 nM for a MSA case (Fig. 1b, Supplementary Fig. 2a, and Supplementary Table 1). In sporadic PD cases, the mean $K_d$ value was $33.5 \pm 17.4$ nM measured across multiple donors using both autoradiography and radiobinding techniques (Supplementary Tables 1 and 2). These data indicate that overall binding affinities were similar across different synucleinopathy cases.

Specific binding of [3H]ACI-12589 to pathological α-syn was further confirmed using post-mortem tissue from different MSA donors (Fig. 1c and Supplementary Figs. 2 and 3). In addition, target engagement was observed in tissues from other synucleinopathies, including idiopathic PD, PD with dementia (PDD), and Lewy Body Variant AD (LBV-AD) (Fig. 1c). No specific signal was observed in tissues from corresponding brain regions from control brains without α-syn pathology. The complete displacement of the [18F]ACI-12589 signal in the presence of an excess of unlabeled ACI-12589 indicates the absence of non-specific binding (NSB) of the ligand to either gray or white matter (Fig. 1c, bottom panel). Quantification of the autoradiography signal as the ratio of the specific signal in disease versus control tissues (Fig. 1d) revealed that [18F]ACI-12589 displayed a 2-3-fold higher signal in PD/PDD and LBV-AD, and a 30-fold higher signal in MSA.

High-resolution autoradiography visualized target engagement on single α-syn inclusions down to a resolution of ~1 μm using autoradiographic emulsion (Fig. 1e). IHC staining of α-syn inclusions (Fig. 1e, top panel) and high-resolution autoradiography with [3H]ACI-12589 on the same sections from a series of human synucleinopathy cases (Fig. 1e, bottom panel) showed an extensive overlap. The weaker autoradiographic signal observed in the MSA case is most likely technical due to the lower physical retention of the photographic emulsion used in this assay on lipid-rich white matter. In contrast to the synucleinopathy samples, [3H]ACI-12589 showed only weak binding to β-amyloid using a brain homogenate from an AD post-mortem brain (Fig. 2a right panel; Supplementary Fig. 4), with a high $K_d$ of approx. 300 nM and a low target occupancy compared to the positive control β-amyloid ligand [3H]PiB ($K_d = 1.4$ nM; Fig. 2a left panel). These data measured in the direct saturation binding experiment with homogenate specifically selected to contain both β-amyloid and tau aggregates (Supplementary Fig. 4), suggests that ACI-12589 has no relevant binding neither to β-amyloid nor to any of the different binding sites of tau. Selectivity over pathological tau was further assessed by high-resolution autoradiography combined with IHC for misfolded tau (Fig. 2b). While the tau tracer [3H]PI-2620 showed a clear autoradiographic signal, no signal was observed with [3H]ACI-12589. Neither was any signal detected with [3H]ACI-12589 to the 4R pathological tau inclusions in PSP tissue (Supplementary Fig. 5). Frontotemporal degeneration (FTD) type C tissue with TDP-43 pathology was also negative for [3H]ACI-12589 binding (Supplementary Fig. 6). Taken together, these data indicate an excellent in vitro selectivity profile of ACI-12589 on α-syn versus pathological β-amyloid, tau, and TDP-43.

To assess whether ACI-12589 can bind to misfolded α-syn in neurodegenerative diseases (NDD) with mixed protein pathologies, high-resolution autoradiography experiments were also conducted on AD tissues containing different levels of α-syn inclusions (Fig. 2c). Interestingly, a clear co-localization between α-syn inclusions by IHC (Fig. 2c, top panel) and [3H]ACI-12589 autoradiographic signal (Fig. 2c, bottom) was observed in α-syn-positive AD tissue. A similar overlap in signal was also detected in α-syn deposits in PSP (Fig. 2c). Labeling of β-amyloid and tau by IHC of sections from AD cases are provided in Supplementary Fig. 7.

Lastly, we evaluated the potential off-target binding of ACI-12589 at a concentration of 1 μM against a panel of more than 100 receptors and enzymes (Supplementary Fig. 8), including monoamine oxidase

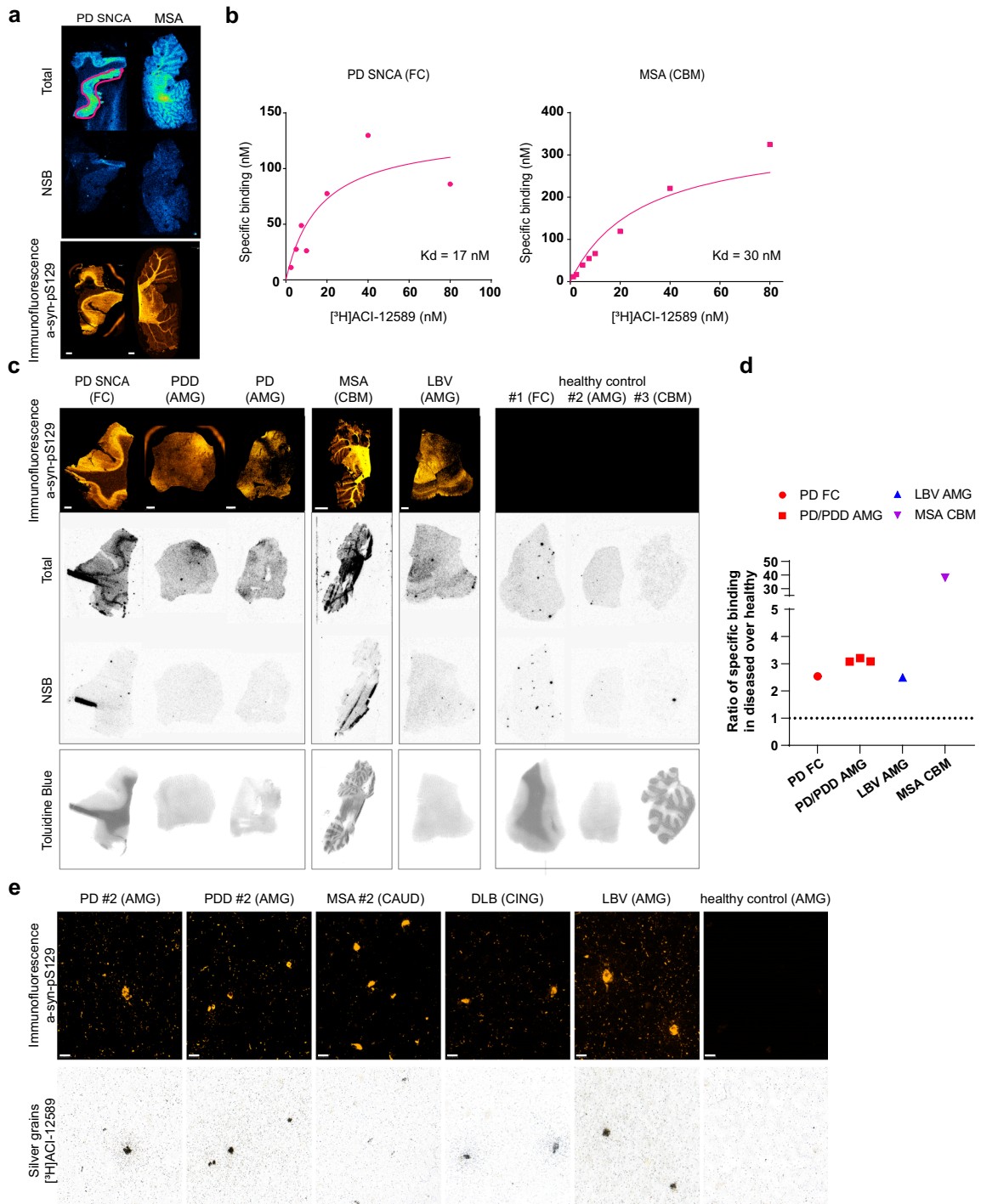

**Fig. 1 | ACI-12589 ex vivo binding properties in α-syn containing brain tissues and controls. a** Autoradiographic detection of [³H]ACI-12589 binding in brain tissue sections from a familial PD (*SNCA*) and an MSA case. Total: total binding (7.5 nM); Non-specific binding (NSB): residual binding in the presence of 5 μM unlabeled ACI-12589. Immunofluorescent staining of adjacent sections with α-syn-pS129. Scale bar, 2 mm. Representative data of at least two independent experiments. **b** Saturation binding studies with [³H]ACI-12589 on human brain tissue sections from the cases shown in (**a**). **c** Autoradiographic detection of [¹⁸F]ACI-12589 binding in brain tissue sections from different α-synucleinopathy cases and healthy controls. Total: total binding (1.7 nM); Non-specific binding: residual binding in the presence of 10 μM unlabeled ACI-12589. Immunofluorescent staining of adjacent sections with α-syn-pS129. Scale bar, 2 mm, except for MSA image 5 mm. Toluidine blue staining differentiates gray versus white matter distributions. Representative data of two independent experiments for PD SNCA and PD. Data from one experiment for the rest of the samples. **d** Ratios of specific binding in PD (red), LBV-AD (blue) and MSA (purple) pathology over the mean, brain region-matched, healthy control values. Dotted line indicates the ratio of 1 corresponding to no difference from healthy controls. Specific binding was calculated as total binding minus NSB for each sample.; PD FC: Frontal cortex (2 replicates of one *SNCA* case); PD/PDD AMG: Amygdala (one PD and 2 PDD cases). Each dot represents one case. When two sections of the same case were analyzed (e.g. SNCA case), the mean value is shown. **e** High-resolution autoradiography with [³H]ACI-12589 (60 nM) in tissue from PD, PDD, MSA, DLB, LBV-AD and a healthy control. Immunofluorescence with α-syn-pS129 antibody (top panels). Accumulation of silver grains on Lewy bodies and neurites on the same section (bottom panels), showing co-labeling α-syn aggregates. Scale bar, 20 μm. Representative data of at least three independent experiments. CBM cerebellum, CAUD caudate, CING cingulate cortex, DLB dementia with Lewy Bodies, LBV-AD Lewy body variant of Alzheimer's disease, PD idiopathic PD, PDD Parkinson's disease with dementia, PD *SNCA* PD due to a *SNCA* G51D mutation.

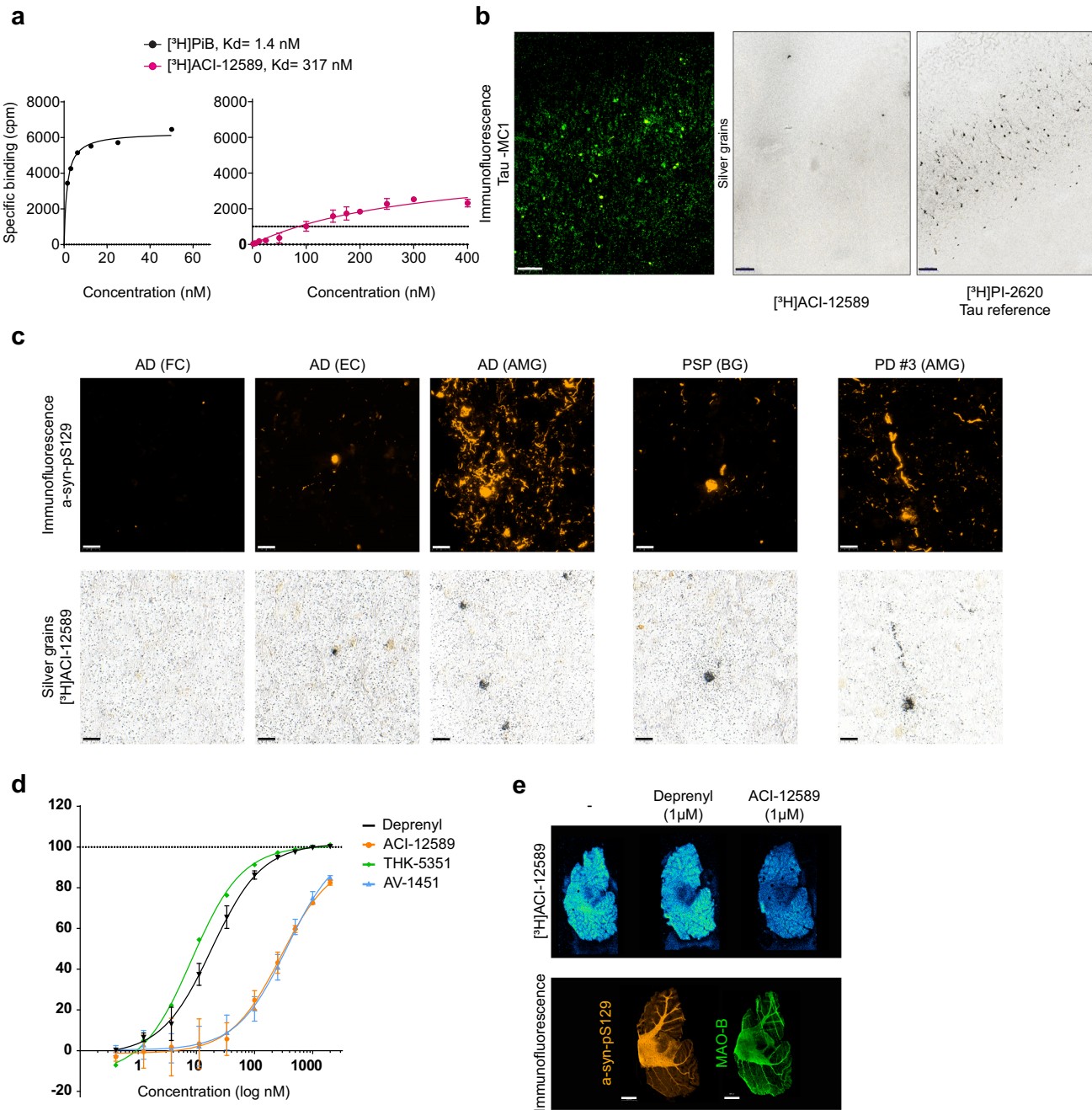

**Fig. 2 | ACI-12589 selectivity over common co-pathologies and off-target binding. a** Assessment of selectivity of [³H]ACI-12589 over β-amyloid and pathological tau aggregates. Saturation binding studies with [³H]ACI-12589 and the β-amyloid ligand [³H]PiB, in AD brain-derived homogenates. Mean ± SD of three independent experiments is shown for [³H]ACI-12589. **b** Immunofluorescence staining with MC1 antibody, labeling tau aggregates, in the entorhinal cortex of an AD patient (left). High-resolution autoradiography using [³H]ACI-12589 (60 nM) in the same tissue section (middle). The tau-binding ligand [³H]PI-2620 (60 nM) was included for reference (right). Scale bars, 200 μm. Representative data of at least three independent experiments. **c** Immunofluorescence with α-syn-pS129 antibody in brain tissue sections from AD, PSP and PD cases showing α-syn pathology (top panels). High-resolution ARG with [³H]ACI-12589 (60 nM) on the same sections, showing co-labeling of α-syn aggregates (bottom panels). Scale bars, 20 μm. Representative data of two independent experiments for AD (FC), PSP and PD#3 and data from one experiment for AD (EC) and AD (AMG). **d** Evaluation of ACI-12589

binding affinity to MAO-B in healthy control-derived brain homogenates. Competition binding studies with [³H]L-deprenyl to determine the inhibitor constant, $K_i$, values for deprenyl (black curve), ACI-12589 (orange curve), AV-1451 (blue curve) and THK-5351 (green curve). When two independent experiments were performed (deprenyl, ACI-12589, AV-1451), Mean ± SEM is shown. **e** Autoradiography with [³H]ACI-12589 (10 nM) in human tissue from the cerebellum of an MSA donor, containing both MAO-B and pathological α-syn aggregates. To assess signal displacement, adjacent sections were incubated with [³H]ACI-12589 in the presence of unlabeled ACI-12589 or Deprenyl at 1 μM concentrations. Autoradiograms for each case are shown. Immunofluorescence staining of adjacent sections with an antibody against phosphorylated α-syn at serine 129 (pSyn-S129) or MAO-B. Scale bar at 5 mm. Representative data of two independent experiments. AD Alzheimer's disease, ARG autoradiography, cpm counts per minute, MAO-B mono-amine oxidase B, MSA multiple system atrophy, PD Parkinson's disease, PSP progressive supranuclear palsy.

(MAO) A, which is a known off-target liability for brain PET tracers[14]. No binding of ACI-12589 to any of these proteins was detected by this method. To also rule out MAO-B binding activity, we conducted displacement assays with the radiolabeled MAO-B inhibitor [³H]deprenyl. As shown in Fig. 2d, ACI-12589 exhibited weak displacement of the MAO-B inhibitor, with a ~100-fold higher $K_i$ than deprenyl itself. We further investigated the possible interference of MAO-B with ACI-12589 binding to α-syn by performing displacement studies in the opposite configuration. These studies were performed in cerebellar tissue sections from a MSA case with both abundant α-syn pathology and elevated MAO-B expression (as shown by IHC in Fig. 2e, bottom panel). The autoradiographic signal obtained with [³H]ACI-12589 was easily displaced by unlabeled ACI-12589 but only minimally displaced by an excess of unlabeled deprenyl (Fig. 2e top panel). This suggests that the specific ACI-12589 signal was largely independent of MAO-B binding. Based on the specific in vitro binding to pathological versus physiological α-syn, as well as the good selectivity over other aggregate-prone proteins and a clean off-target profile, [¹⁸F]ACI-12589 was selected for an initial clinical evaluation.

## Participants

To evaluate the performance of [¹⁸F]ACI-12589 in vivo a total of 42 participants were recruited. Participants consisted of eight healthy controls, and 23 participants with synucleinopathies, comprising eight with PD (including two with duplications of the *SNCA* gene), two with DLB, and 13 with MSA. Eleven participants with other NDDs were also recruited, comprising five AD, three PSP, and three with hereditary ataxias (two with Friedreich ataxia and one with a mutation in the *SAMD9L* gene). Participant demographics are presented in Table 1. The initial 25 scans were performed as dynamic scans with arterial blood sampling ($n = 22$) for input function and metabolite analysis. Remaining scans were performed as static scans 60–90 min post injection.

## In vivo kinetic analysis

Kinetic analysis indicated a rapid brain uptake and a rapid washout of radioactive [¹⁸F]ACI-12589 from regions without specific binding and retention of the tracer in distinct brain structures, including the cerebellar white matter in MSA subjects (Fig. 3a and Supplementary Fig. 9a). The radiotracer showed a good stability over the time of the scan with $66.7 \pm 8.4\%$ of the parent fraction remaining at 90 min (Fig. 3b and Supplementary Fig. 9b). Mean whole blood and plasma input functions are shown in Fig. 3c. Tracer kinetics showed good fits to Ichise Multilinear analysis MA1 and Logan graphical analysis, and the resulting distribution volumes ($V_T$) derived with MA1 and Logan plot were in close agreement. $V_T$ values derived by Logan graphical analysis using arterial input indicated no difference in tracer uptake between diagnostic groups in the occipital cortex or cerebellar gray matter, indicating that

these regions could potentially be used as reference regions in simplified analyses (Supplementary Fig. 10a, b). Outcomes derived using arterial input functions ($V_T$ values derived from Logan graphical analysis) and reference tissues $BP_{ND}$ (SRTM or Logan reference) were well correlated ($R = 0.87$; Fig. 3d and Supplementary Fig. 10c). The retention of the tracer in the occipital cortex and cerebellar gray matter were highly correlated, using the Logan reference model (Fig. 3e and Supplementary Fig. 10d). Moreover, simplified analyses using standardized uptake value ratio with a cerebellar cortex reference region ($SUVR_{cer}$) showed linear relationships to the Logan reference binding potential ($BP_{ND}$) values, indicating that $SUVR_{cer}$ values derived from 30-min static scans in the 60–90-min time interval after injection can be used to estimate [¹⁸F]ACI-12589 tracer retention (Fig. 3f and Supplementary Fig. 10e).

## In vivo retention in synucleinopathies

Representative images of transverse sections through the cerebellum and at the level of the lenticular nuclei in control, PD, DLB and MSA participants are shown in Fig. 4a, b (additional images for all controls and MSA participants are presented in Supplementary Figs. 11 and 12). In participants with MSA, there was a significantly increased retention of the [¹⁸F]ACI-12589 radiotracer in the cerebellar white matter (Fig. 4c, d) compared to healthy controls (MSA: $1.68 \pm 0.44$, Controls $1.01 \pm 0.08$; Wilcoxon rank sum test $p < 0.0001$) and participants with Lewy body disease (LBD = PD and DLB $1.01 \pm 0.10$; Wilcoxon rank sum test $p < 0.0001$) with no overlap between MSA-C and the LBD groups. The retention in the cerebellar white matter was higher in participants with MSA-C compared to MSA-P (MSA-C $1.88 \pm 0.43$, MSA-P $1.37 \pm 0.20$, Wilcoxon rank sum test $p = 0.02$). Similar results were obtained for the middle cerebellar peduncles (Supplementary Fig. 13a). Moreover, the MSA-P participants exhibited an increased retention of [¹⁸F]ACI-12589 in the lentiform nuclei that was not seen in MSA-C participants without parkinsonian features (Supplementary Fig. 13b; MSA-P vs controls, LBD and MSA-C, Wilcoxon rank sum test all $p < 0.05$).

Overall, however, the retention of [¹⁸F]ACI-12589 in the basal ganglia was more difficult to interpret because of the variable levels of retention between subjects, as well as visible levels of basal ganglial signal in some participants with PD, but also in a subset of healthy controls (Supplementary Fig. 13b). A tendency for a lower retention was observed in the basal ganglia of PD participants on MAO-B inhibitors, raising the possibility that a fraction of the [¹⁸F]ACI-12589 binding in the basal ganglia could be MAO-B-related (Supplementary Fig. 11). To elucidate whether the MSA-specific signal could be explained by off-target binding to MAO-B, six of the MSA-participants (5 MSA-C and 1 MSA-P) were rescanned after blocking MAO-B with 10 mg selegiline daily for 6 days (Fig. 5). No reduction of the cerebellar white matter signal with selegiline pre-treatment (SUVR change: $2 \pm 10\%$, Wilcoxon signed rank test $p = 0.18$) was observed, indicating

## Table 1 | Demographics

| | Controls | PD | DLB | MSA | PSP | AD | Ataxia |
|---|---|---|---|---|---|---|---|
| *N* | 8 | 8 | 2 | 13 | 3 | 5 | 3 |
| Age | $63 \pm 11$ | $68 \pm 6$ | $81 \pm 1$ | $61 \pm 8$ | $72 \pm 9$ | $69 \pm 4$ | $54 \pm 14$ |
| Sex (M/F) | 5/3 | 7/1 | 2/0 | 7/6 | 3/0 | 4/1 | 2/1 |
| Injected dose (MBq) | $314 \pm 39$ | $308 \pm 56$ | $289 \pm 1$ | $297 \pm 13$ | $298 \pm 8$ | $296 \pm 5$ | $267 \pm 67$ |
| UMSARS I + II | – | – | – | $53 \pm 23$ | – | – | – |
| UPDRS-III | – | $65 \pm 16$ | – | – | – | – | – |
| Hoehn&Yahr | – | $2 \pm 0$ | $2 \pm 0$ | $3.5 \pm 1.2$ | $4 \pm 0$ | – | – |
| Number with dynamic data (number with arterial input function) | 8 (7) | 6 (5) | 2 (2) | 9 (8) | – | – | – |

Demographic data of participants. Ataxia includes three participants with hereditary ataxia, two with Friedreich Ataxia and one with a mutation in the *SAMD9L* gene.
*AD* Alzheimer's dementia, *DLB* dementia with lewy bodies, *F* female, *M* male, *MSA* multiple system atrophy, *PD* Parkinson's disease, *PSP* progressive supranuclear palsy, *UMSARS* unified MSA rating scale, *UPDRS-III* unified PD rating scale, motor part.

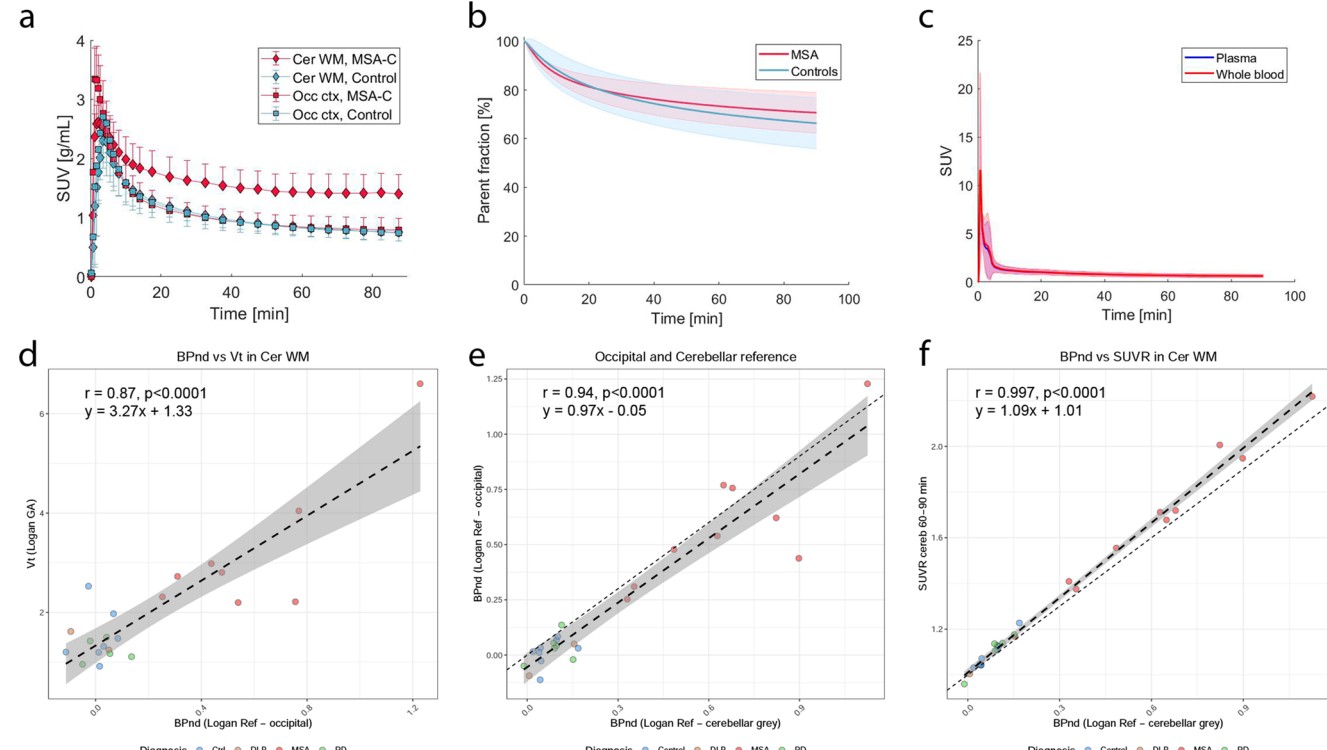

**Fig. 3 | Kinetic modeling of [¹⁸F]ACI-12589 in vivo. a** Time activity curves for cerebellar white matter (diamonds) and occipital cortex (squares) in MSA (red, $n = 6$) and control participants (blue, $n = 7$) over the course of the PET scan (mean ± SD). **b** Parent fractions for MSA (red, $n = 6$) and control subjects (blue; $n = 7$; mean ± SD). **c** Whole blood (red) and plasma (blue) input functions. **d** Comparison of $V_T$s in the cerebellar white matter derived from blood input Logan graphical analysis and Logan reference $BP_{ND}$ values in the cerebellar white matter using an occipital reference region. $R$-value 0.87 (95% C.I. [0.71–0.95]) derived using Pearson's correlation ($t = 7.9643$, df = 20, $p$ value = 1.2e−07). **e** Comparison of Logan reference $BP_{ND}$ values in the cerebellar white matter with occipital and cerebellar reference regions. The cerebellar reference region shows slightly higher $BP_{ND}$ values in comparison to the occipital reference. $R$-value 0.94 (95% C.I.

[0.87–0.97]) derived using Pearson's correlation ($t = 13.401$, df = 23, $p$ value = 2.4e −12). **f** Comparison of SUVR$_{cer}$ values in the cerebellar white matter in the 60–90 min time interval with Logan reference $BP_{ND}$ values, showing a linear relationship. $R$-value 0.997 (95% C.I. [0.992–0.998]) derived using Pearson's correlation ($t = 57.186$, df = 23, $p$ value < 2.2e−16). The error bands in (**d**–**f**) represent the 95% C.I. of the dashed linear regression line. $BP_{ND}$ non-displaceable binding potential, Cer WM cerebellar white matter, DLB dementia with Lewy bodies, Logan GA Logan graphical analysis model, Logan Ref Logan reference tissue model, MSA multiple system atrophy, MSA-C multiple system atrophy with a cerebellar phenotype, PD Parkinson's disease, SUV standardized uptake value, SUVR$_{cer}$ Standardized uptake value ratio, with a cerebellar gray matter reference region, $V_T$ volume of distribution.

that this signal was not attributable to MAO-B. In the lentiform nuclei, a considerable inter-subject variability was noted, with an overall 17 ± 16% (Wilcoxon signed rank test $p = 0.47$) decrease in the SUVR signal after selegiline administration. Although this suggests limited off-target binding to MAO-B in the basal ganglia overall, further analyses will be required to understand whether this may be condition- and/or subregion-specific.

We did not detect an increased retention of the tracer in the brainstem or in the cerebral cortex of sporadic PD or DLB participants (Supplementary Fig. 13c, d, g, h). In the two participants with familial PD due to an *SNCA* duplication (one very early symptomatic and one with parkinsonism and mild cognitive impairment [MCI]), retention signals were observed above control levels in the pons as well as in the upper range of controls in the midbrain and cerebral cortex (Supplementary Fig. 14). Interestingly, the cortical retention was more pronounced in the case with MCI, where a more widespread α-syn pathology might be expected based on clinical symptoms.

Taken together, these data indicate that the signal observed in the cerebellar white matter and peduncles differentiates MSA cases from controls and other synucleinopathies and, based on its spatial distribution, identifies an underlying α-syn pathology.

**In vivo retention in other neurodegenerative diseases**
As a next step, the retention of [¹⁸F]ACI-12589 in other NDDs was investigated to evaluate whether its binding was unique to MSA. We

included participants with PSP (as it is a key differential diagnostic entity to MSA), hereditary cerebellar ataxias (as they also affect the cerebellum; Fig. 6a) and AD (as AD cause widespread neurodegeneration; Fig. 6b). In PSP, we detected some uptake in subregions of the basal ganglia overlapping with the retention in MSA (Fig. 6a and Supplementary Figs. 11 and 13b). In the ataxia participants, retention was observed in the cerebellar white matter and cerebellar peduncles, the levels of which were within the lower range of signals observed in MSA patients (Fig. 6c, d). In AD cases, retention of [¹⁸F]ACI-12589 was observed in the cerebral cortex, with lower levels in the cerebellum. Interestingly, the regions of increased cortical retention were correlated to, albeit not completely overlapping with, areas of tau tracer ([¹⁸F]RO948) uptake (Fig. 6b and Supplementary Fig. 15a–d). In contrast, only a weak correlation between the retention of [¹⁸F]ACI-12589 with β-amyloid ([¹⁸F]flutemetamol) PET was observed (Fig. 6b and Supplementary Fig. 15e, f).

## Discussion
A PET ligand to detect α-syn pathology in vivo has long been an unmet clinical need. The major obstacle to this achievement has been the identification of ligands capable of selectively detecting α-syn inclusions, which are present at a much lower density than other targets, such as pathological β-amyloid or tau[10]. We report here the preclinical and initial clinical characterization of ACI-12589, which has demonstrated: (i) that it is possible to visualize brain α-syn pathology in

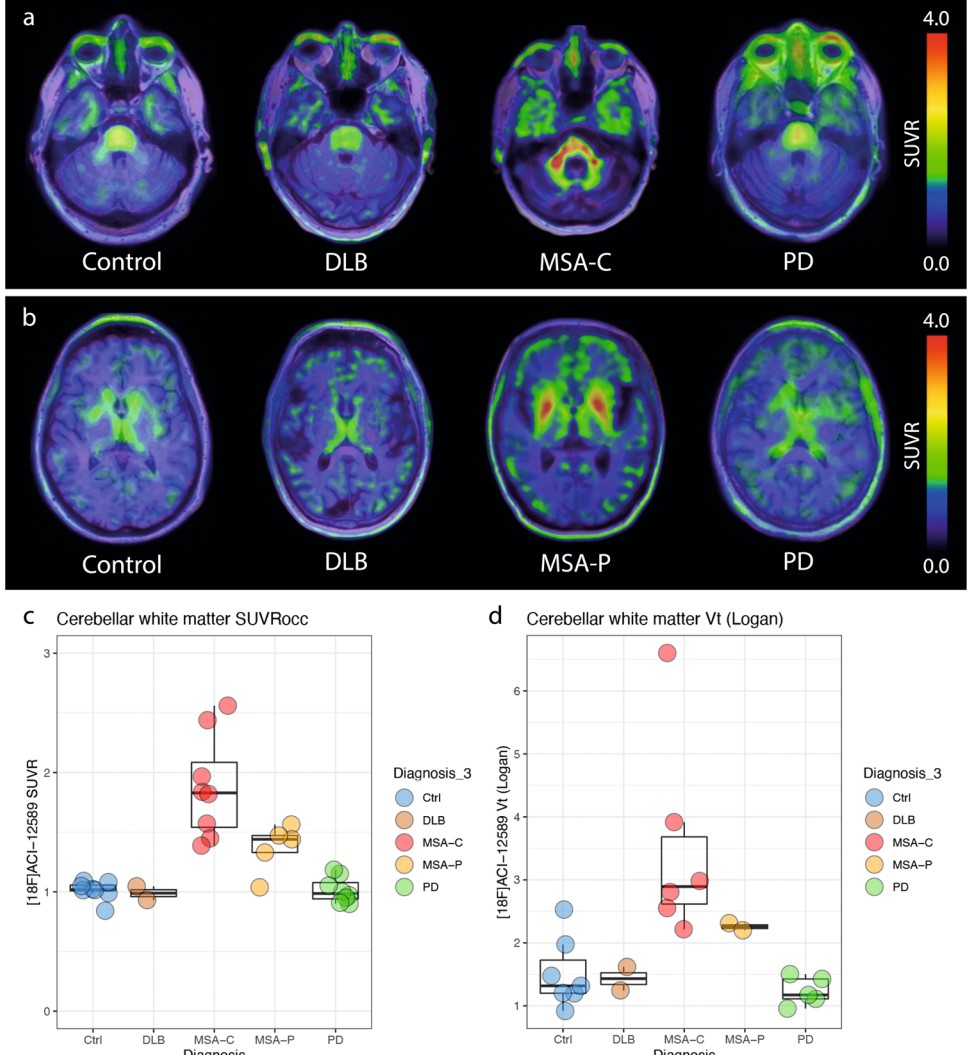

**Fig. 4 | [¹⁸F]ACI-12589 PET in participants with α-synucleinopathies.**
**a** Transversal images at the level of the middle cerebellar peduncles in a control participant, and patients with DLB, MSA-C and PD. **b** Transversal images at the level of the basal ganglia in a control participant, and patients with DLB, MSA-P and PD. SUVR images for (**a**, **b**) have been created using occipital cortex as a reference region. **c** SUVR values in the cerebellar white matter in the different disease groups (Ctrl $n = 8$, DLB $n = 2$, MSA-C $n = 8$, MSA-P $n = 5$, PD $n = 8$). **d** $V_T$ values derived from Logan graphical analysis modeling in the cerebellar white matter in the different disease groups with available blood and dynamic PET data. Boxplots show median, IQR (box) and whiskers (Q1 − 1.5*IQR/Q3 + 1.5*IQR or minimum/maximum value, outliers not included) (Ctrl $n = 7$, DLB $n = 2$, MSA-C $n = 6$, MSA-P $n = 2$, PD $n = 5$). DLB dementia with Lewy bodies, MSA-C multiple system atrophy with a cerebellar phenotype, MSA-P multiple system atrophy with a parkinsonian phenotype, occ occipital cortex reference region, PD Parkinson's disease, SUVR standardized uptake value ratio, $V_T$ volume of distribution.

patients by PET and, (ii) that this signal can differentiate MSA cases from controls and other NDD cases.

The in vitro properties of ACI-12589 indicate that the tracer binds specifically to α-syn with $K_d$ values of $33.5 \pm 17.4$ nM, while showing good selectivity versus β-amyloid and tau. Ex vivo, target engagement was demonstrated to α-syn inclusions present both as primary pathology, such as in MSA, PD, DLB and PDD tissues, but also as co-pathology, such as in AD and PSP tissues. Finally, in patients, a high PET signal for [¹⁸F]ACI-12589 was observed in the cerebellar white matter and middle cerebellar peduncles in participants with both MSA dominated by cerebellar ataxia (MSA-C) and MSA dominated by parkinsonism (MSA-P). The retention in cerebellar structures was higher in participants with cerebellar and brain stem symptoms (MSA-C) as compared to MSA-P. Tracer uptake was also observed in the lentiform nuclei of participants with MSA-P, indicative of a basal ganglia involvement. The uptake thus correlated well with the expected distribution of α-syn pathology in both MSA subtypes[6,15]. The present results

are congruent with a short report describing differential binding patterns of the α-syn radiotracer [¹⁸F]SPAL-T-06 in a limited study of four participants, three with MSA and one control[16]. In contrast to the previous report, the present study describes a large body of work that extends our initial observations presented in conference proceedings[17] and establishes the use of α-syn targeting tracers, such as [¹⁸F]ACI-12589, to demonstrate a specific and reproducible pattern of pathology in patients with both MSA-P and MSA-C subtypes.

[¹⁸F]ACI-12589 retention in the cerebellar white matter and cerebellar peduncles clearly distinguished participants with MSA from controls and participants with PD or DLB. No region-specific retention was observed in PD or DLB. There are several potential explanations for the lack of clear retention of [¹⁸F]ACI-12589 in PD and DLB. One likely reason is the lower density of α-syn pathology in these disorders compared to MSA[18]. This notion is supported by elevated [¹⁸F]ACI-12589 retention found in the familial PD cases due to α-syn (*SNCA*) gene duplication, who are expected to exhibit a more dense and widespread α-syn pathology

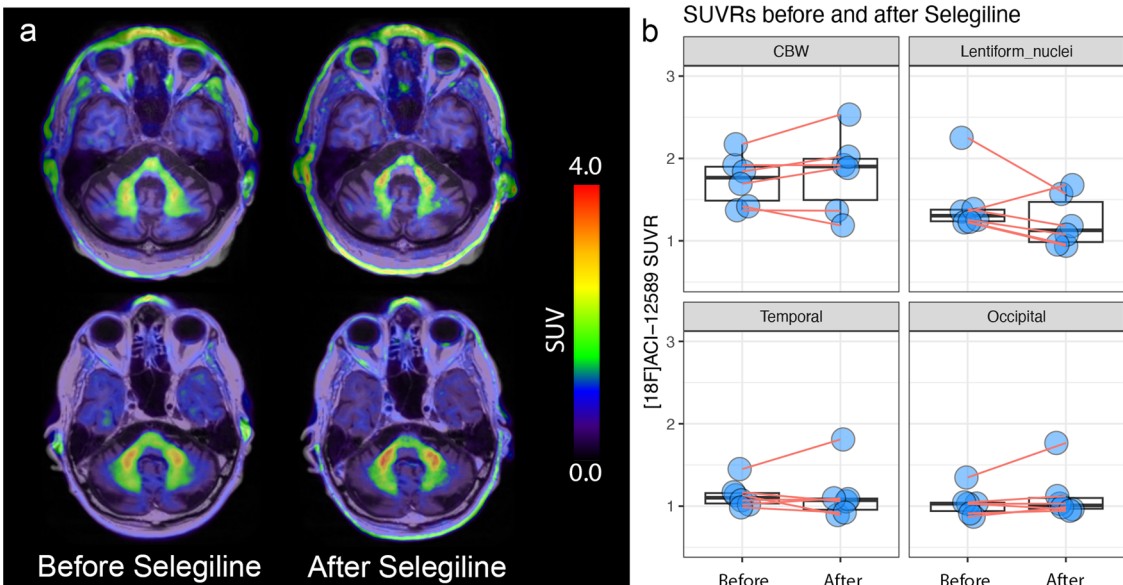

**Fig. 5 | [¹⁸F]ACI-12589 retention before and after Selegiline. a** shows SUV images from two participants with MSA-C before (left) and after (right) treatment for 6 days with 10 mg Selegiline. **b** shows changes in SUVR before and after treatment (*n* = 6) in Cerebellar white matter (CBW), Lentiform Nuclei (summed bilateral putamen and globus pallidus), Temporal cortex and Occipital cortex. Boxplots show median, IQR (box) and whiskers (Q1 − 1.5*IQR/Q3 + 1.5*IQR or minimum/maximum value, outliers not included). CBW cerebellar white matter, SUV standardized uptake value, SUVR standardized uptake value ratio.

compared to idiopathic PD cases[18,19]. Interestingly, one of the monogenic PD cases with *SNCA* duplication who exhibited cognitive impairment, also displayed some cortical retention of [¹⁸F]ACI-12589 which is consistent with the expected pathologic burden underlying this phenotype. Nonetheless, studies in additional non-MSA synucleinopathy cases with different expected α-syn loads and distribution patterns will be needed to draw any firm conclusions. Another possible explanation for the low binding in PD/DLB is potential differences in the conformation of the α-syn aggregates in idiopathic PD/DLB compared to MSA. Such differences have been recently described at a molecular level by cryo-EM[7–9], and are also reported based on the recently developed RT-QuIC seeding amplification assays[8,9]. Additionally, differences in α-syn isoform expression and post-translational modifications between the different synucleinopathies[20] could potentially explain differences in [¹⁸F]ACI-12589 retention.

When comparing MSA cases with participants with other NDDs, such as hereditary ataxias, PSP and AD, some tracer retention was observed in disease-affected brain areas even in the non-MSA cases. The reasons for this retention are not fully clear and off-target binding to another neurodegenerative process or the presence of α-syn co-pathologies are potential and not mutually exclusive explanations. The presence of multiple co-pathologies is a known feature in NDDs[21], and α-syn deposits are frequently found in AD, but also reported in PSP, among others[22–24]. The premise that the observed PET signal could in part be α-syn-related is further supported by data showing that ACI-12589 can bind to α-syn co-pathology in AD and PSP tissues ex vivo. Although ACI-12589 has a clean off-target profile in vitro, we cannot rule out that some of the disease-related signal in vivo represent off-target binding. We have, however, considered some of the most likely contributors associated with neuroinflammatory or neurodegenerative processes. An established in vivo paradigm of blocking MAO-B with selegiline[21] showed no interference with [¹⁸F]ACI-12589 retention in the cerebellar white matter in MSA, speaking against a meaningful contribution of MAO-B binding to the observed PET signal in this region. In vitro data also characterized possible MAO-B binding as limited. Nonetheless, we cannot rule out a contribution of MAO-B to the retention of [¹⁸F]ACI-12589, particularly in high MAO-B-expressing areas such as the basal ganglia. We found a colocalization and

significant correlation between [¹⁸F]ACI-12589 and [¹⁸F]RO948 (a tau-PET ligand) in vivo in AD. Similarly in PSP, [¹⁸F]ACI-12589 retention matched the expected distribution of tau pathology. As no target engagement of ACI-12589 to tau neurofibrillary tangles was reported ex vivo, it is unlikely that binding to pathological tau is the primary source for this PET signal. Possibly, the co-localization of [¹⁸F]ACI-12589 and [¹⁸F]RO948 signals might reflect off-target binding to a neurodegenerative process downstream of tau. The exact nature of the binding of this PET ligand in different NDDs will be a focus of future work.

As the first scans in humans, the aim of the present work was to understand the utility of [¹⁸F]ACI-12589 as a PET tracer for α-syn-mediated NDDs, which led to the discovery of its capacity to significantly discriminate MSA cases. The main limitation of this study is the relatively low number of participants, particularly for synucleinopathies other than MSA and other NDDs. Therefore, results concerning binding in these diseases should be considered preliminary. Similarly, the low and variable binding in sporadic PD merits further in-depth studies with larger patient cohorts. Strengths of the study are the extensive preclinical characterization of ACI-12589 and the in vitro data supporting α-syn specificity and selectivity, the dynamic scanning with arterial blood sampling and kinetic modeling of the in vivo performance of the radiotracer, as well as the relatively large cohort of MSA participants recruited.

In conclusion, the present results indicate that [¹⁸F]ACI-12589 could be used to improve the diagnostic work-up of MSA, leading to an earlier diagnosis. Furthermore, these findings indicate that α-syn tracers could allow detection of drug target engagement in vivo in clinical trials of novel α-syn targeting therapies.

## Methods
### Human brain samples
Post-mortem frozen tissue blocks from different brain regions of control donors and donors with confirmed α-syn pathology were acquired from the Netherlands Brain Bank (NBB; Netherlands Institute for Neuroscience, Amsterdam (open access www.brainbank.nl)), Queen Square Brain Bank (QSBB), Banner Health Institute Brain & Tissue Bank, and commercial providers (Tissue Solutions Ltd.,

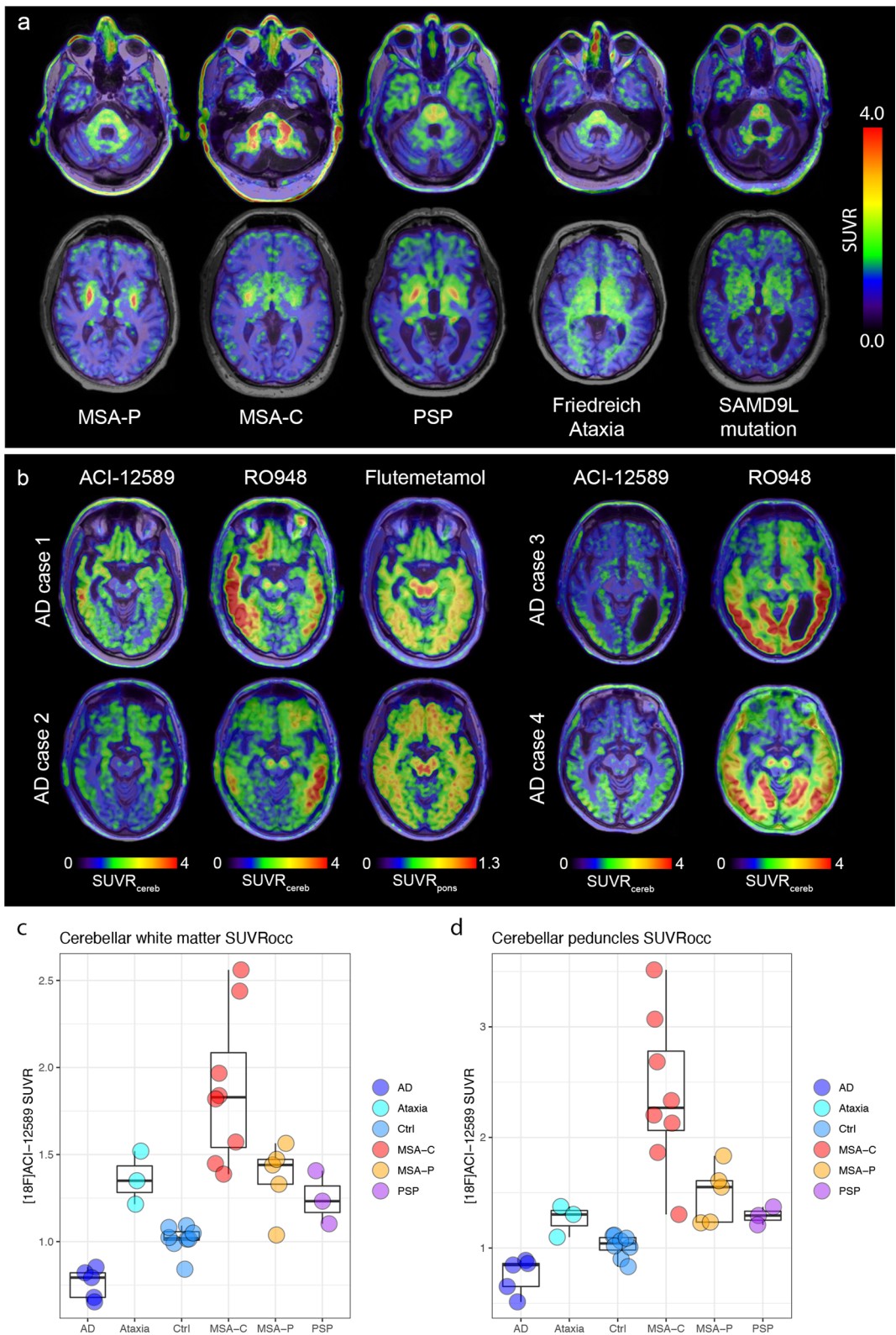

Glasgow, UK). All tissues have been collected from donors for or from whom a written informed consent for a brain autopsy and the use of the material and clinical information for research purposes had been obtained. The QSBB is supported by the Reta Lila Weston Institute of Neurological Studies, UCL Queen Square Institute of Neurology. Tissues included one familial Parkinson's disease (PD) case with a mutation (G51D) in the Synuclein Alpha (SNCA) gene, one sporadic PD case,

two PD dementia cases, three MSA cases, one Lewy body variant Alzheimer's disease (LBV-AD), one AD and one PSP case. Demographics of the donors are summarized and neuropathology reports in Supplementary Tables 4 and 5. Frozen tissue blocks from the frontal cortex, entorhinal cortex, amygdala, basal ganglia and cerebellum of patients and control cases were processed using a cryotome to generate 10 µm sections that were mounted on glass slides. Sections were kept at

**Fig. 6 | [18F]ACI-12589 retention in other neurodegenerative disorders.**
**a** transversal images at the level of the middle cerebellar peduncles (upper row) and the basal ganglia (bottom row) in representative patients with MSA-C, MSA-P, PSP, Friedreich Ataxia and cerebellar ataxia due to a *SAMD9L* mutation. SUVR images in (**a**) have been created using occipital cortex as reference region. **b** transversal images at the level of the substantia nigra in patients with AD using [18F]ACI-12589, [18F]RO948 (tau), and [18F]Flutemetamol (β-amyloid) as indicated above. SUVR images were generated using a cerebellar (ACI-12589 and RO948) or pons (Flutemetamol) reference region. **c** SUVR$_{occ}$ values in the cerebellar white matter in the different disease groups. **d** SUVR$_{occ}$ values in the middle cerebellar

peduncles in the different disease groups. Boxplots show median, IQR (box) and whiskers (Q1 − 1.5*IQR/Q3 + 1.5*IQR or minimum/maximum value, outliers not included). *N*−numbers for (**c**, **d**): AD $n = 5$, Ataxia $n = 3$, Ctrl $n = 8$, MSA-C $n = 8$, MSA-P $n = 5$, PSP $n = 3$. AD Alzheimer's disease, Ataxia cerebellar ataxias (Friedreich Ataxia and cerebellar ataxia due to a *SAMD9L* mutation), cereb cerebellar gray matter reference region, Ctrl control subjects, MSA-C multiple system atrophy with a cerebellar phenotype, MSA-P multiple system atrophy with a parkinsonian phenotype, PSP progressive supranuclear palsy, SUVR$_{occ}$ SUVR with an occipital cortex reference region, SUVR standardized uptake value ratio.

−80 °C until use. The presence or absence of pathological α-syn aggregates in tissue sections from α-synucleinopathy or control donors, respectively, was confirmed by immunostaining. Sections were counterstained with Toluidine blue (TB) to differentiate the gray versus white matter.

## Immunofluorescence staining
Sections were fixed for 15 min at 4 °C with 4% formaldehyde (Sigma, 252549) and washed three times 5 min with 1× PBS (Dulbecco's phosphate buffered saline, Sigma D1408) at room temperature (RT). Next, sections were blocked and permeabilized in blocking buffer (PBS, 10% NGS, 0.25% Triton X-100) for 1 h at RT and incubated overnight at 4 °C with an antibody specific for phosphoserine 129 α-syn (α-syn-pS129, 1:500, Abcam 51253), or the conformation-dependent anti-Tau antibody (MC1, 1:200), or anti-pTDP-43 pS409/410 (Biolegend, 829901, 1:500), or an antibody against MAO-B (Thermofisher, PA5-28338, 1:500). The following day, sections were washed three times 5 min with 1× PBS before incubation with a secondary, AlexaFluor647-labeled goat-anti-rabbit antibody (Abcam, ab150079, 1:500) or goat-anti-mouse antibody (Jackson ImmunoResearch, 115-605-166, 1:500) or AlexaFluor633-labeled goat-anti-rat antibody (Invitrogen, A-21094, 1:500). Following incubation with secondary antibody the sections were washed three times in PBS. For image acquisition, sections were mounted using ProLong Gold Antifade reagent (Invitrogen P36930) and imaged on a Panoramic250 Slide Scanner (3DHistech) with a ×20 objective.

## [18F]ACI-12589 and [3H]ACI-12589 autoradiography
[18F]ACI-12589 was synthesized as described below. Frozen human sections were first equilibrated for 30 min in assay binding buffer [50 mM Tris-HCl (pH 7.4), 120 mM NaCl, 5 mM KCl, 2 mM CaCl$_2$, 1 mM MgCl$_2$, 0.1% BSA] and then incubated with 1.8 nM [18F]ACI-12589 (specific activity 11.1 GBq/μmol) in assay binding buffer for 60 min at RT or with 10 nM [3H]ACI-12589 or increasing concentrations of [3H]ACI-12589 in the range of 1.25 nM to 80 nM (specific activity 51.1 Ci/mmol) in assay binding buffer for 120 min at RT. To determine non-specific binding (NSB), adjacent brain sections were incubated with [18F]ACI-12589 mixed with 10 μM of unlabeled ACI-12589 or with [3H]ACI-12589 mixed with 5 μM of unlabeled ACI-12589. To assess potential competition between [3H]ACI-12589 and Deprenyl for binding to MAO-B, 10 nM [3H]ACI-12589 was mixed with 1 μM of unlabeled Deprenyl or ACI-12589, respectively.

For [18F]ACI-12589 autoradiography, slides were washed three times for 5 min in washing buffer [50 mM Tris-HCl (pH 7.4) at 4 °C] and then, dipped briefly in distilled water. For [3H]ACI-12589 autoradiography, slides were washed sequentially in washing buffer for 1 min; twice in ice-cold PBS for 1 min; and in washing buffer for 1 min.

For [18F]ACI-12589 autoradiography, slides were allowed to air-dry before being placed under Phosphor imaging screens (Fujifilm Plate BAS-TR2025, Fujifilm, Tokyo, Japan) in imaging cassettes for 55 min. Imaging screens were scanned using a phosphor imaging system (Fujifilm BAS-5000 phosphor imager, Tokyo, Japan) and resulting images were analyzed using Multi Gauge 3.2 phosphor imager software (Fujifilm, Tokyo, Japan) for ROI delimitation and quantification and GraphPad Prism v7 for analyses. Specific binding was determined by

subtracting the non-specific signal from the total signal. When artifacts were present, appearing as randomly placed black dots in some of the samples, they were excluded from the analyses. The ratio of specific signal in diseased versus control tissues was determined by dividing the specific signal in the diseased tissue over the average (if several cases were tested) specific signal in the control tissue for each brain region.

For [3H]ACI-12589 autoradiography, slides were allowed to air-dry before being exposed and scanned in a real-time autoradiography system (BeaQuant instrument, ai4R) for 2 h. ROI delimitation and quantification of signal was performed by using the image analysis software Beamage (ai4R). Specific binding was determined by subtracting the non-specific signal (NSB) from the total signal. $K_d$ (dissociation constant) values were calculated in GraphPad Prism v7 by applying a nonlinear regression curve fit using a one site, specific binding model.

## Assessment of target engagement of [3H]ACI-12589 by high-resolution micro-autoradiography
The protocol was adapted from Marquie et al.[25]. Sections were incubated with 60 nM of [3H]ACI-12589 or [3H]PI-2620 for one hour at room temperature (RT). Sections were then washed as follows: One time in ice-cold 50 mM Tris-HCl pH 7.4 buffer for 1 min, two times in 70% ice-cold ethanol for 1 min, one time in ice-cold 50 mM Tris-HCl pH 7.4 buffer for 1 min and finally rinsed briefly in ice-cold distilled water. Sections were subsequently dried and then exposed to Ilford Nuclear Emulsion Type K5 (Agar Scientific, AGP9281) in a light-proof slide storage box. The sections were developed by immersing them successively in the following solutions: (1) Ilford Phenisol Developer (Agar Scientific, AGP9106), (2) Ilfostop solution (Agar Scientific, AGP9104), (3) Ilford Hypam Fixer (Agar Scientific, AGP9183) and finally rinsed with H$_2$O. Immunostaining was also performed on the same section using an antibody specific for phosphorylated serine at amino acid 129 α-synuclein (α-syn-pS129, Abcam 51253), as described above. For image acquisition, sections were mounted using ProLong Gold Antifade reagent (Invitrogen P36930) and imaged on a Panoramic150 Slide Scanner (3DHistech) with a ×20 objective.

## Participants and PET-imaging
Thirteen MSA, three PD−two of which carried a duplication in the α-syn (*SNCA*) gene[26], two DLB, three PSP, five AD, two with Friedreich Ataxia[27] and one with a mutation in the *SAMD9L* gene[28], as well as three normal control participants were recruited at the Memory Clinic and the Department of Neurology, Skåne University Hospital, Sweden (Sept 2021−May 2022) whereas five PD patients and five normal controls were recruited by Invicro, LLC (New Haven, Connecticut, USA). Both males and females were recruited to the study. 30 participants were assigned a male sex and 12 a female sex. Participants were not asked for gender identity or self-reported sex. No sex or gender-based sub-analysis of the results was performed due to the low number of participants. MSA participants fulfilled criteria for probable or possible MSA according to the 2008 Gilman-criteria[29]. LBD participants fulfilled National Institute of Neurological Disorders and Stroke (NINDS) diagnostic criteria for PD[30], or the McKeith-criteria for DLB[2]. AD

participants fulfilled DSM-V criteria for dementia due to AD and PSP participants 2017 Movement disorder society criteria for PSP[31]. Written informed consent was obtained from all participants prior to entering the study. The study was approved by the Swedish Ethical Review Authority. Participants were compensated for travel expenses.

Twenty-five participants underwent dynamic scans 0–90 min after injection of $305 \pm 39$ MBq [18F]ACI-12589 on a GE Discovery MI digital PET/CT scanner (Lund) or a ECAT EXACT HR+ scanner (Invicro, LLC). Arterial blood samples for plasma input function and metabolite analysis were collected in 22/25 participants with dynamic scans. Image data was collected in LIST-mode and were reconstructed into $6 \times 30$ s, $4 \times 60$ s, $4 \times 120$ s, $15 \times 300$ s time frames. The remaining 19 participants underwent static scans ($6 \times 5$ min frames) in the 60–90 min time interval post injection. Images were reconstructed using a VPFX-S (ordered subset expectation maximization combined with corrections for time-of-flight and point spread function) algorithm, with 6 iterations and 17 subsets, 3 mm smoothing in plane smoothing, standard Z filter, and a 25.6-cm field of view ($256 \times 256$ matrix)[12]. PET-images were motion corrected and co-registered to T1-MPRAGE MRI images using Pmod 3.7 (Pmod Ltd, Zürich, Switzerland). Regions of interest (ROIs) were defined using the Automated Anatomical Labeling (AAL) atlas as implemented in Pmod 3.7. The middle cerebellar peduncles were manually delineated on the MRI (Supplementary Fig. 16). The co-registration between MRI and PET images was checked manually and the ROIs in subcortical structures in the brain were manually adjusted for all subjects (using the MRI data only) to exclude cerebrospinal fluid from the ROIs. Kinetic analysis was performed using Pmod 3.7 and using an in-house developed kinetic pipeline as described in the Supplementary Information.

### Radiosynthesis, metabolite and radioactivity analysis
The radiosynthesis of [18F]ACI-12589, quality control, blood sampling and metabolite analysis is detailed in the Supplementary Methods, in Supplementary Table 3 and Supplementary Fig. 17.

### Statistics
All statistical tests were two-tailed with a significance level of 0.05. Kruskal–Wallis and Wilcoxon rank sum tests were used for group comparisons. Wilcoxon signed rank tests were used for paired comparisons. Correlations were calculated as Pearson correlations. All analyses were performed using the R programming language (v 4.0.3).

### Reporting summary
Further information on research design is available in the Nature Portfolio Reporting Summary linked to this article.

## Data availability
Anonymized data will be shared by request from a qualified academic investigator for the sole purpose of replicating procedures and results presented in the article and if data transfer is in agreement with EU legislation on the general data protection regulation and decisions by the Swedish Ethical Review Authority, which should be regulated in a material transfer agreement. Source data are provided with this paper.

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

## Acknowledgements

Work at Skåne University Hospital and Lund University was supported by the Michael J Fox Foundation (MJFF-019169), the Swedish Research Council (2022-00775), ERA PerMed (ERAPERMED2021-184), the Knut and Alice Wallenberg foundation (2017-0383), the Strategic Research Area MultiPark (Multidisciplinary Research in Parkinson's disease) at Lund University, the Swedish Alzheimer Foundation (AF-980907), the Swedish Brain Foundation (FO2021-0293), The Parkinson foundation of Sweden (1294/20, 1412/22), the Cure Alzheimer's fund, the Konung Gustaf V:s och Drottning Victorias Frimurarestiftelse, the Skåne University Hospital Foundation (2020-O000028), Regionalt Forskningsstöd (2022-1259) and the Swedish federal government under the ALF agreement (2020-Projekt0110, 2022-Projekt0080). Work at AC Immune was supported by by the Michael J Fox Foundation (MJFF-010312, MJFF-019055, MJFF-019169). Human tissue was kindly provided by the Netherlands Brain Bank (NBB); the Queen Square Brain Bank and Banner Sun Health Research Institute Brain and Body Donation Program. The Queen Square Brain Bank is supported by the Reta Lila Weston Institute of Neurological Studies, UCL Queen Square Institute of Neurology. The Banner Sun Health Research Institute Brain and Body Donation Program is supported by the National Institute of Neurological Disorders and Stroke (U24 NS072026 National Brain and Tissue Resources for Parkinson's Disease and Related Disorders), the National Institute on Aging (P30 AG19610 Arizona Alzheimer's Disease Core Center), the Arizona Biomedical Research Commission (contracts 4001,0011,05–901 and 1001 to the Arizona Parkison's Disease Consortium), and the Micheal J. Fox Foundation for Parkinsons's Research. We also thank Professor Andrea Varrone for his contribution to the autoradiography work performed at the Karolinska Institute.

## Author contributions

R.S., F.C., J.S., and O.H. contributed to the design and implementation of the research and to the analysis of the results. JM contributed to the invention of ACI-12589. I.K.D., T.O., K.B., and E.T.L. contributed to radiosynthesis process development and optimization. R.S., M.S., O.H., and C.S. contributed to acquisition and analysis of clinical data. E.V. contributed to the design of the preclinical research. P.M.-A. and C.H. contributed to the autoradiography experiment with [$^{18}$F]ACI-12589. R.S., A. Puschmann, P.S., M.A., D.S.R., and S.H. contributed to recruitment of the subjects. V.H., T.T., E.S., and J.J. contributed to clinical operations. R.S., F.C., M.S., E.V., R.L.-C., M.K.-V., A. Pfeifer, and O.H. drafted the manuscript. All authors revised the manuscript.

## Funding

## Competing interests

R.S. has received a speaker fee from Roche. F.C., E.V., J.M., T.T., V.H., I.K.D., R.L.-C., A. Pfeifer, M.K.-V., and J.S. are employees of AC Immune SA. M.S. is an employee of Antaros Medical. A Puschmann receives reimbursement from Elsevier for his work as Associate Editor for Parkinsonism and Related Disorders. P.S. has received consultancy/speaker fees from Abbvie, Amylyx, Lundbeck, and Takeda. C.S. and D.S.R. are employees of Invicro, LLC. O.H. has acquired research support (for the institution) from ADx, AVID Radiopharmaceuticals, Biogen, Eli Lilly, Eisai, Fujirebio, GE Healthcare, Pfizer, and Roche. In the past 2 years, he has received consultancy/speaker fees from AC Immune, Amylyx, Alzpath, BioArctic, Biogen, Cerveau, Eisai, Eli Lilly, Fujirebio, Genentech, Merck, Novartis, Novo Nordisk, Roche, Sanofi, and Siemens. T.O., J.J., M.A., P.M.-A., C.H., E.S., S.H., K.B., and E.T.L. report no disclosures.

## Additional information

[1]Clinical Memory Research Unit, Department of Clinical Sciences, Malmö, Lund University, Lund, Sweden. [2]Department of Neurology, Skåne University Hospital, Lund, Sweden. [3]AC Immune SA, EPFL Innovation Park, Building B, 1015 Lausanne, Switzerland. [4]Antaros Medical, Mölndal, Sweden. [5]Neurobiology Research Unit, Copenhagen University Hospital, Copenhagen, Denmark. [6]Department of Radiation Physics, Skånes University Hospital, Lund, Sweden. [7]Neurology, Department of Clinical Sciences Lund, Lund University, Lund, Sweden. [8]SciLifeLab National Research Infrastructure, Lund University, Lund, Sweden. [9]Department of Clinical Physiology and Nuclear Medicine, Skåne University Hospital, Lund, Sweden. [10]Department of Neurology, Academic Specialist Center, Karolinska University Hospital, Stockholm, Sweden. [11]Invicro, LLC, New Haven, CT, USA. [12]Clinical Neuroscience, PET Division, Karolinska Institute, Stockholm, Sweden. [13]Memory Clinic, Skåne University Hospital, Lund, Sweden. [14]Department of Biomedical Sciences, University of Antwerp, Antwerp, Belgium. [15]These authors contributed equally: Ruben Smith, Francesca Capotosti. ✉e-mail: johannes.streffer@uantwerpen.be; Oskar.Hansson@med.lu.se

