## [Peer Review File · Nature Communications]

The α -synuclein PET tracer [18F] ACI-12589 distinguishes multiple system atrophy from other neurodegenerative diseasesREVIEWER COMMENTS

Reviewer #1 (Remarks to the Author):

This paper describes a series of in vitro characterization studies and first-in-human imaging studies of a PET radiotracer for imaging alpha synuclein aggregates in the synucleinopathies such as Parkinson's disease (PD) and multiple system atrophy (MSA). This has been an extremely important area of research over the past decade, and little progress has been made in this topic. The authors have presented some compelling data in their studies with [18F]ACI-12589, which appears to be able to image alpha synuclein in MSA but not PD in in vivo PET imaging studies. These results are contradictory to what was expected based on autoradiography studies in postmortem brain specimens of PD and MSA brain. The development of a PET radiotracer for imaging alpha synuclein, even if it only works in MSA and not PD, represents a significant advance in the study of synucleinopathies. This paper should be of widespread interest to the PET community doing research studies in neurodegeneration. However, there are significant deficiencies in the presentation of the data described in this paper that must be corrected before it can be accepted for publication. These deficiencies are presented below.

1. The authors need to show the structure of the precursor for the radiolabeling studies, the structure of [18F]ACI-12589, and conditions of the radiolabeling reaction in a Figure or Scheme. This is standard practice for the publication of a new radiotracer in primary scientific literature, a practice that every academic site must adhere to. The authors have failed to disclose this information at their scientific presentations in the past. They must now disclose this information. This paper should not be published otherwise.
2. Abstract. The authors must state that the PET imaging studies in PD subjects did not reveal any specific uptake of the radiotracer. As written, the abstract gives the impression that this radiotracer works in PD but the in vivo imaging data proved otherwise.
3. The authors have not adequately characterized the binding of their radiotracer for tau. They looked at binding in AD (mixed 3R/4R tau) but not in the 4R tauopathies. Furthermore, it is well known that there are multiple binding sites for tau radioligands; therefore, one

must screen an alpha synuclein radioligand for tau binding for tau present in AD (3R/4R tau) and the different 4R tauopathies (CBD, PSP). They must also use different tau radioligands as described in ACS Chem Neurosci. 2021; 12: 596–602. The fact that [18F]ACI-12589 overlaps with the tau radiotracer [18F]RO948 in AD and PSP subjects suggests that it may bind to tau. The in vivo K_d of this radioligand may have a higher affinity than the 317 nM observed in the in vitro binding studies in AD brain. Finally, the authors need to do more detailed studies characterizing the binding of their radioligand to 4R tau in PSP.

4. Page 13, lines 292-296: The rationale for the failure of their radiotracer to work in PD as described in this section is speculation and must be backed up with experimental data. To make a claim as such is not appropriate. The authors can easily calculate B_{max} values from Scatchard transformations of the in vitro binding data shown in the Supplementary information.

5. The neuropathology report for the human tissue sections in Figures 1 and 2, Suppl Figure 1 and 2 (i.e., pathology scores for Abeta, tau and Asyn) must be given.

6. The data in Supplementary Table 2 is confusing. How can this ligand have a negative % inhibition of 30% for some of the CNS targets in a radioligand binding assay? This suggests that their ligand increases the binding of the radioligand used in the assay.

Reviewer #2 (Remarks to the Author):

This is the first comprehensive report of a novel alpha synuclein (AS) PET tracer ACI-12589 that visualizes brain pathology in MSA compared to controls and other neurodegenerative disorders. High retention levels were found in cerebellar white matter in MSA-C/P and in lentiform nuclei in MSA-P. In contrast there was no region specific retention in PD/DLB consistent with a lower density or strain related properties of AS in Lewy body disorders. Interestingly, AS duplication cases did show ACI-12589 binding.

Some retention was seen in non-MSA cases suggesting off-target binding and/or co-AS pathologies. However, blocking with selegiline showed no interference with ACI-12589 binding in MSA cerebellar white matter and AD cortex. There was matching of ACI-12589

and tau binding in vivo in AD and PSP. In summary the study discovered a novel AS PET imaging tracer for MSA. This can be used as diagnostic as well as a progression biomarker indicating target engagement in clinical trials. The study is not definitive since it is limited by the low number of non-MSA AS disorders as well as low binding in Lewy body disorders.

Reviewer #3 (Remarks to the Author):

The manuscript by Smith et al. describes the in vitro and first in human evaluation of the [18F] ACI-12589 PET tracer targeting alpha synuclein. A PET tracer for alpha synuclein is a long waited medical need that will have a significant impact on the understanding of the disease progression and drug development. This work is very novel and of great importance to the scientific community. The study is well designed and of a high quality. The tracer has been well characterized for selectivity and affinity in vitro however the number cases used per disease is rather low and could be beneficial to do the evaluation with a bigger number of tissue to get more robust results. The human imaging study is well performed and it is nice to see that the tracer is able to image alpha synuclein at least in MSA. It is however a pity that the authors did not disclose the structure of the compound

Minor comments:

- 1- In figure 1, it would be more informative to have additional immunostaining with more zoomed images for the alpha synuclein staining showing more in detail the pathology. This could be included in the supplementary material
- 2- In figure 1d, is the dots on the graph showing the mean from different cases or just one staining? It is more informative to have multiple cases per disease and show each case as one dot on the graph to give an impression on the spread.
- 3- In figure 1E, MSA#2 case, in the silver stain data, it seems that the binding of the [18F] ACI-12589 is very low on alpha synuclein aggregates which is not in accordance with the rest? Can you explain?
- 4- In figure 2C, did you perform amyloid beta and tau staining on the AD cases? What is the extent of the pathology in the figures you show?
- 5- Was the binding assay of [18F] ACI-12589 on PD and human tissue performed in n=1 (no

SD on graph) and on one single tissue? That would not be robust enough to report a K_d . Has a binding assay been performed on PFFs? In addition, blocking with cold ACI-12589 in autoradiography and binding assay seems not optimal as you can observe that it also block non-specific binding. K_d claims should be carefully reported.

6- Line 199, " $R^2=0.76$ " no in accordance with the figure 3d or supp 4c

7- In Sup figure 7 when looking at the AD cases the SUVR in many brain regions are lower compared to control or other disease. Is there any explanation for this?

8- In figure 6 the order of AD cases presentation is not in sequence.

9- It is interesting to see a difference of [18F] ACI-12589 uptake between the MCA-C and P. Is it because there is a difference in the level of pathology? would be interesting to add a discuss part about the difference observed.

REVIEWER COMMENTS

Reviewer #1 (Remarks to the Author):

This paper describes a series of in vitro characterization studies and first-in-human imaging studies of a PET radiotracer for imaging alpha synuclein aggregates in the synucleinopathies such as Parkinson's disease (PD) and multiple system atrophy (MSA). This has been an extremely important area of research over the past decade, and little progress has been made in this topic. The authors have presented some compelling data in their studies with [18F]ACI-12589, which appears to be able to image alpha synuclein in MSA but not PD in in vivo PET imaging studies. These results are contradictory to what was expected based on autoradiography studies in postmortem brain specimens of PD and MSA brain. The development of a PET radiotracer for imaging alpha synuclein, even if it only works in MSA and not PD, represents a significant advance in the study of synucleinopathies. This paper should be of widespread interest to the PET community doing research studies in neurodegeneration. However, there are significant deficiencies in the presentation of the data described in this paper that must be corrected before it can be accepted for publication. These deficiencies are presented below.

Response: Dear reviewer, thank you very much for finding interest in our findings and pointing out the flaws in the analyses/presentation. We have addressed your concerns in a point-by-point manner below, and hope that you will find the added information suitable and sufficient for publication.

1. The authors need to show the structure of the precursor for the radiolabeling studies, the structure of [18F]ACI-12589, and conditions of the radiolabeling reaction in a Figure or Scheme. This is standard practice for the publication of a new radiotracer in primary scientific literature, a practice that every academic site must adhere to. The authors have failed to disclose this information at their scientific presentations in the past. They must now disclose this information. This paper should not be published otherwise.

Response:

The structure of the chemical precursor ACI-15051, and the radiolabeled compound [18F]ACI-12589 have been added to the manuscript as Supplementary Figure 1, with a reference in the Results section (page 6, lines 116-117): "The structure of the tracer [18F]ACI-12589 and its precursor, ACI-15051, as well as the radiolabeling reaction, are shown in Suppl. Fig. 1." The details of the manufacturing process of [18F]ACI-12589 PET tracer are provided in the supplementary information section (Supplementary information, page 8), with a reference from the Methods (page 16, line 374).

2. Abstract. The authors must state that the PET imaging studies in PD subjects did not reveal any specific uptake of the radiotracer. As written, the abstract gives the impression that this radiotracer works in PD but the in vivo imaging data proved otherwise.

Response:

We agree with the reviewer that the wording in the abstract could give the impression that the radiotracer showed *in vivo* binding also in PD. To clarify this, we have now rephrased the abstract: "In this first-in-human study, [18F]ACI-12589 demonstrated clear binding in the cerebellar white matter and middle cerebellar peduncles of MSA patients, regions known to be highly affected by α -synuclein pathology, *but showed limited binding in PD.*" (added text in italics). Page 3, line 53

3. The authors have not adequately characterized the binding of their radiotracer for tau. They looked at binding in AD (mixed 3R/4R tau) but not in the 4R tauopathies. Furthermore, it is well known that there are multiple binding sites for tau radioligands; therefore, one must screen an alpha synuclein radioligand for tau binding for tau present in AD (3R/4R tau) and the different 4R tauopathies (CBD, PSP). They must also use different tau radioligands as described in ACS

Chem Neurosci. 2021; 12: 596–602. The fact that [18F]ACI-12589 overlaps with the tau radiotracer [18F]RO948 in AD and PSP subjects suggests that it may bind to tau. The in vivo K_d of this radioligand may have a higher affinity than the 317 nM observed in the in vitro binding studies in AD brain. Finally, the authors need to do more detailed studies characterizing the binding of their radioligand to 4R tau in PSP.

Response:

Acknowledging the complexity of assessing Tau selectivity for each possible binding site, we have taken the approach of simultaneously evaluating the binding of [3H]ACI-12589 to all the existing binding sites performing a direct saturation binding experiment. For this experiment, brain homogenate from an AD case was selected to contain both pathological β -amyloid and Tau. The weak affinity measured to such homogenate (Fig. 2a) confirms therefore that ACI-12589 has no relevant binding neither to β -amyloid nor to any of the different binding sites of Tau. For completeness, the biochemical characterization of this homogenate confirming the presence of β -amyloid and aggregated Tau has been now added to the supplementary information (Supplementary Figure 4) and the text in the manuscript has been rephrased to better clarify this concept. (new text in *italics*) Page 7 (lines 147-153): “In contrast to the synucleinopathy samples, [3H]ACI-12589 showed only weak binding to β -amyloid using a brain homogenate from an AD post-mortem brain (Fig. 2a right panel; *Suppl. Fig 4*), with a high K_d of approx. 300 nM and a low target occupancy compared to the positive control β -amyloid ligand [3H]PiB ($K_d = 1.4$ nM; Fig. 2a left panel). *These data measured in the direct saturation binding experiment with homogenate specifically selected to contain both β -amyloid and Tau aggregates (Suppl. Fig. 4), suggests that ACI-12589 has no relevant binding neither to β -amyloid nor to any of the different binding sites of tau*”

In addition, we have further investigated a possible interaction of ACI-12589 with pathological Tau from 4R Tauopathies by performing high-resolution autoradiography experiments on brain tissues from 2 distinct PSP donors. Similarly to what previously shown for 3R/4R Tau in AD (Fig. 2b), [3H]ACI-12589 does not show any specific binding to the 4R pathological Tau inclusions in PSP (Supplementary Figure 5). These results further confirm the selectivity of ACI-12589 against the different forms of pathological Tau. A sentence has been added to the results section of the manuscript (added text in *italics*. Page 7, lines 154-158): “Selectivity over pathological tau was *further* assessed by high-resolution autoradiography combined with IHC for misfolded tau (Fig. 2b). While the tau tracer [3H]PI-2620 showed a clear autoradiographic signal, no signal was observed with [3H]ACI-12589. *Neither was any signal detected with [3H]ACI-12589 to the 4R pathological tau inclusions in PSP tissue (Supplementary Figure 5).*”.

4. Page 13, lines 292-296: The rationale for the failure of their radiotracer to work in PD as described in this section is speculation and must be backed up with experimental data. To make a claim as such is not appropriate. The authors can easily calculate B_{max} values from Scatchard transformations of the in vitro binding data shown in the Supplementary information.

Response:

The claim about a possibly lower target density of α -syn pathology in PD versus MSA was supported by initial observations from saturation binding studies performed in PD and MSA tissue sections and reported in Fig. 1b. Despite the limited number of cases analysed, these results indicate a higher B_{max} in MSA versus PD tissue in alignment with previous reports on the abundance of α -syn aggregates in PD and MSA (Tong J, *et al. Brain* **133**, 172-188 (2010)).

For better clarity, B_{max} values for the cases reported in Fig. 1b are now shown in the Table below and has been added as Supplementary information (Supplementary Table 1). In addition, B_{max} values for 2 idiopathic PD cases (PDD) have now been added. Together these data indicate a progressively decreasing B_{max} from MSA to familial PD to idiopathic PD cases, in line with what was observed in vivo.

The supplementary results have been referenced in the manuscript main text, page 6, lines 124-129: "Saturation binding experiments performed by autoradiography showed dissociation constants (K_d) of 17 nM for a familial PD case and 28 nM for a MSA case (Fig. 1b; Suppl. Fig 2a and Suppl Table 1). *In sporadic PD cases, the mean K_d value was 33.5 ± 17.4 nM measured across multiple donors using both autoradiography and radiobinding techniques (Suppl. Table 1 and 2). These data indicate that overall binding affinities were similar across different synucleinopathy cases.*".

5. The neuropathology report for the human tissue sections in Figures 1 and 2, Suppl Figure 1 and 2 (i.e., pathology scores for Abeta, tau and Asyn) must be given.

Response:

Neuropathology reports for the human tissue sections used in the manuscript are now provided in the Supplementary Information (Supplementary Table 6, pages 37-38).

6. The data in Supplementary Table 2 is confusing. How can this ligand have a negative % inhibition of 30% for some of the CNS targets in a radioligand binding assay? This suggests that their ligand increases the binding of the radioligand used in the assay.

Response:

The results reported in Supplementary Table 2 (now supplementary table 3) have been performed by Eurofins, a contract research organization specialized in performing off-target screening. According to the criteria provided by Eurofins for data interpretation, only a result showing an inhibition higher than 50% is considered to represent a significant effect of the tested compound. Results showing an inhibition lower than 25%-30% are not considered significant and mostly attributable to variability of the signal around the control level. Similarly, it is reported that low to moderate negative values have no real meaning and are attributable to variability of the signal around the control level. High negative values ($\geq 50\%$) that are sometimes obtained are generally attributable to nonspecific effects of the test compounds in the assays or, on rare occasion, they could suggest an allosteric effect of the test compound.

Specifically, for the reported results, no effect $\geq 50\%$ was reported at the tested concentration (1 μM) and therefore it was concluded that no off-target binding potentially relevant *in vivo* was identified.

This has been further clarified in the legend of the supplementary table 3, page 21. "In vitro off-target binding assay. *In each assay, only results showing effects larger than 50% are considered relevant. Effects in the 25%-30% are not considered significant and mostly attributable to variability of the signal around the control level.*"

Reviewer #2 (Remarks to the Author):

This is the first comprehensive report of a novel alpha synuclein (AS) PET tracer ACI-12589 that visualizes brain pathology in MSA compared to controls and other neurodegenerative disorders. High retention levels were found in cerebellar white matter in MSA-C/P and in lentiform nuclei in MSA-P. In contrast there was no region specific retention in PD/DLB consistent with a lower density or strain related properties of AS in Lewy body disorders. Interestingly, AS duplication cases did show ACI-12589 binding.

Some retention was seen in non-MSA cases suggesting off-target binding and/or co-AS pathologies. However, blocking with selegiline showed no interference with ACI-12589 binding in MSA cerebellar white matter and AD cortex. There was matching of ACI-12589 and tau binding in vivo in AD and PSP. In summary the study discovered a novel AS PET imaging tracer for MSA. This can be used as diagnostic as well as a progression biomarker indicating target engagement in clinical trials. The study is not definitive since it is limited by the low number of non-MSA AS disorders as well as low binding in Lewy body disorders.

Response:

We thank the reviewer for finding interest in our manuscript and for the accurate summary of the results of the paper. We agree with the reviewer that the results are still preliminary with regards to

the binding in synucleinopathies other than MSA where the binding intensity is more limited and where the sample size is lower. We have acknowledged this in the limitation section at the end of the discussion (page 15, lines 351-355): “*The main limitation of this study is the relatively low number of participants, particularly for synucleinopathies other than MSA and other NDDs. Therefore, results concerning binding in these diseases should be considered preliminary. Similarly, the low and variable binding in sporadic PD merits further in-depth studies with larger patient cohorts.*”

Reviewer #3 (Remarks to the Author):

The manuscript by Smith et al. describes the *in vitro* and first in human evaluation of the [18F] ACI-12589 PET tracer targeting alpha synuclein. A PET tracer for alpha synuclein is a long waited medical need that will have a significant impact on the understanding of the disease progression and drug development. This work is very novel and of great importance to the scientific community. The study is well designed and of a high quality. The tracer has been well characterized for selectivity and affinity *in vitro* however the number cases used per disease is rather low and could be beneficial to do the evaluation with a bigger number of tissue to get more robust results. The human imaging study is well performed and it is nice to see that the tracer is able to image alpha synuclein at least in MSA. It is however a pity that the authors did not disclose the structure of the compound

Response:

We thank the reviewer for the helpful comments and appreciation of our work.

In line with the request also from reviewer #1 (main concern) the structure of the compound has been included as Supplementary Figure 1 and details of radiosynthesis are specified in the supplementary methods (Supplementary Methods, page 8).

We have also added information to the *in vitro* data set:

- 1) Assessing the different binding properties in MSA, hereditary PD and idiopathic PD (Supplementary Table 1)
- 2) Adding more details from the Neuropathology reports to the supplementary information (Supplementary Table 6)
- 3) Further characterizing the potential off-target binding to tau in PSP, *in vitro* (Supplementary Figure 5).

Minor comments:

1- In figure 1, it would be more informative to have additional immunostaining with more zoomed images for the alpha synuclein staining showing more in detail the pathology. This could be included in the supplementary material

Response:

As requested, zoomed images of Fig. 1C have been added in the Supplementary information (Supplementary Figure 3). Please note that the tissue is fresh frozen for the autoradiography and therefore the structural detail may be less distinct compared to fixed tissue slices.

2- In figure 1d, is the dots on the graph showing the mean from different cases or just one staining? It is more informative to have multiple cases per disease and show each case as one dot on the graph to give an impression on the spread.

Response:

In Fig. 1d, each dot represents one donor. For example, for the PD/PDD amygdala group, the values for one PD and two PDD donors are shown. If several sections per donor were analyzed, the mean value was shown for each case. The figure legend has now been updated to clarify this point. (Page 20, lines 461-463): “*Each dot represents one case. When two sections of the same case were analyzed (e.g. SNCA case), the mean value is shown*”.

3- In figure 1E, MSA#2 case, in the silver stain data, it seems that the binding of the [18F] ACI-12589 is very low on alpha synuclein aggregates which is no in accordance with the rest? Can you explain?

Response:

In Fig. 1e, we show high-resolution autoradiography data with [³H]ACI-12589 across different α -synucleinopathy cases. Indeed, as opposed to what we observe by classical autoradiography (Fig. 1a, Fig. 1b), the silver grain signal in the caudate of an MSA donor appears less intense compared to PD or other α -synucleinopathy cases. This is due to the lower physical retention of the photographic emulsion used for the high-resolution autoradiography on lipid-rich white matter tissues such as the typical ones from MSA cases. We have observed this technical problem across different sections and ligands tested. This explanation is now added in the Results section to avoid any confusion (Page 7, lines 145-147): *“The weaker autoradiographic signal observed in the MSA case is most likely technical due to the lower physical retention of the photographic emulsion used in this assay on lipid-rich white matter”*.

4- In figure 2C, did you perform amyloid beta and tau staining on the AD cases? What is the extend of the pathology in the figures you show?

Response:

As shown in Supplementary Table 6, added in the Supplementary Information section, we now provide the histopathological characterization of each tissue donor, either provided directly from the neuropathologist of the Brain Bank from where we received the tissue or following in house characterization, by immunofluorescence labeling for b-amyloid Tau and a-syn. The AD donor used in Fig. 2c, was characterized as Braak VI and in-house staining shows that the samples contain large amounts of Ab and Tau pathological aggregates. Images of such staining have been added to the Supplementary Information (Supplementary Figure 7), and a sentence referring to the information included in the results section (page 8, lines 167-168): *“Labeling of b-amyloid and tau by immunohistochemistry of sections from AD cases are provided in Suppl. Fig 7.”*

5- Was the binding assay of [18F] ACI-12589 on PD and human tissue performed in n=1 (no SD on graph) and on one single tissue? That would not be robust enough to report a kd. Has a binding assay been performed on PFFs? In addition, blocking with cold ACI-12589 in autoradiography and binding assay seems not optimal as you can observe that it also block non-specific binding. Kd claims should be carefully reported.

Response:

Saturation binding studies performed on human brain tissue slides, as shown in Fig. 1b were performed in one PD and one MSA donor. In these experiments, each dot represents specific binding (defined as total minus non-specific binding) for the indicated concentration, therefore 2 brain sections are used per each concentration tested. Unfortunately, this significantly limits the number of donors and the experimental replicates that can be provided. Nevertheless, additional Kd values for two PDD cases have been now added (Supplementary Table 1) for a total now of four synucleinopathy cases. Additionally, the Kd data set has been extended with additional values measured from saturation binding experiments run on brain homogenates from PD/PDD cases. As shown in (Supplementary Table 2), multiple donors and brain regions were used for those experiments and number for experimental replicates as well as the 95% confidence interval is provided for each Kd value. This information is now added to the Supplementary Information and to the results section, page 6, lines 126-127: *“In the sporadic PD cases, the mean K_d value was 33.5 ± 17.4 nM measured across multiple donors using both autoradiography and radiobinding techniques (Suppl. Table 1 and 2)”*.

Concerning the PFF, ACI-12589 was not tested on any recombinant a-syn aggregates as the relevance of the conformation of such aggregates for the human pathology and therefore their translational value is limited.

Regarding the specificity of displaceable signal, we are presenting side-by-side with the autoradiograms, the immunofluorescence staining of pSyn aggregates across the entire (adjacent) section. This allows for the reader to evaluate the correlation of the strength and localization of the autoradiography signal with the density and distribution of the a-syn pathology. We have now added some zoomed-in images for Fig. 1a to better illustrate the correlation of the autoradiography signal with the presence of pathological a-syn aggregates. Finally, in Suppl. Fig. 2a, we show the total and non-specific binding curves from the saturation binding studies, in which one can observe the expected linearity of the non-specific signal.

6- Line 199, "R²=0.76" no in accordance with the figure 3d or supp 4c

Response:

We are sorry if this was inconsistent, the R² value reported in the results was the squared R-value reported in Figure 3d (r=0.87). For consistency with Figure 3d, we have changed the wording in the results (now page 10, line 212 to "R=0.87").

7- In Sup figure 7 when looking at the AD cases the SUVR in many brain regions are lower compared to control or other disease. Is there any explanation for this?

Response:

This is an effect of the occipital cortex being used as the reference region. We opted to use the occipital cortex as a reference since it provides a reliable reference region for MSA and for comparisons to the other synucleinopathies. The cerebellar grey matter reference region provides similar results but since there is an atrophy of the cerebellar cortex in MSA there is also a risk that atrophy and partial volume effects with CSF results in falsely too low values in the reference region and an overestimation of the results in MSA if a cerebellar reference is used. To avoid this issue we chose to use the occipital cortex.

However, for AD the occipital reference region contains some binding and therefore the results in other regions are lower in comparison to other diseases and controls.

8- In figure 6 the order of AD cases presentation is not in sequence.

Response:

Thank you for pointing this out. The participants were named after order of recruitment to the study, but the ordering does not make sense in the figure presentation. The images have been renamed.

9- It is interesting to see a difference of [18F] ACI-12589 uptake between the MSA-C and P. Is it because there is a difference in the level of pathology? would be interesting to add a discuss part about the difference observed.

Response:

We think that this is the most likely interpretation of the findings (i.e., more symptoms in regions that are more affected by a more widespread pathology). We have extended the discussion to further discuss the differences between phenotypes. The discussion now reads (added text in *italics*), page 13, lines 291-298: "Finally, in patients, a high PET signal for [18F]ACI-12589 was observed in cerebellar white matter and middle cerebellar peduncles in participants with both MSA dominated by cerebellar ataxia (MSA-C) and MSA dominated by parkinsonism (MSA-P). *The retention in cerebellar structures was higher in participants with cerebellar and brain stem symptoms (MSA-C) as compared to MSA-P. Tracer uptake was also observed in the lentiform nuclei of participants with MSA-P, indicative of a basal ganglia involvement. The uptake thus correlated well with the expected distribution of a-syn pathology in both MSA subtypes.*"

REVIEWERS' COMMENTS

Reviewer #1 (Remarks to the Author):

The authors have made the recommended changes, and the manuscript has been significantly improved. It is now acceptable for publication.

Reviewer #3 (Remarks to the Author):

The reviewer is satisfied with the reply to the comments and the modifications in the manuscript